# A Cohort Study of Free Light Chain Ratio in Combination with Serum Protein Electrophoresis as a First-Line Test in General Practice

**DOI:** 10.3390/cancers14122930

**Published:** 2022-06-14

**Authors:** Birgitte Sandfeld-Paulsen, Ninna Aggerholm-Pedersen, Mie Hessellund Samson, Holger Jon Møller

**Affiliations:** 1Department of Clinical Biochemistry, Aarhus University Hospital, 8200 Aarhus, Denmark; miesamso@rm.dk (M.H.S.); holgmoel@rm.dk (H.J.M.); 2Department of Oncology, Aarhus University Hospital, 8200 Aarhus, Denmark; ninnpede@rm.dk; 3Department of Experimental Oncology, Aarhus University Hospital, 8200 Aarhus, Denmark

**Keywords:** serum-free light chain (sFLC) ratio, Multiple Myeloma, general practice, diagnostic work-up

## Abstract

**Simple Summary:**

Multiple Myeloma (MM) can be a diagnostic challenge as it often presents with unspecific symptoms in patients in general practice. Serum-free light chain (sFLC) ratio is suggested to replace urine protein electrophoresis (UPE) in the diagnostic work-up of myeloma. We aimed to investigate the performance of the sFLC ratio in general practice (GP) compared to UPE in a low prevalence cohort of 13,210 patients from general practice. We found that sFLC ratio performs in line with UPE; however, we observed a pronounced number of false-positive tests. Therefore, local instrument-dependent adjustment of reference ranges/decision limits should be considered to avoid an unnecessarily high number of false-positive tests.

**Abstract:**

Multiple Myeloma (MM) often present with unspecific symptoms, which can lead to diagnostic delay. Serum-free light chain (sFLC) ratio is suggested to replace urine protein electrophoresis (UPE) in the diagnostic work-up of myeloma. We aimed to investigate the performance of the sFLC-ratio in general practice (GP) compared to UPE, just as we explored different sFLC-ratio cut-offs’ influence on diagnostic values. In a cohort of 13,210 patients from GP measures of sFLC-ratio, serum protein electrophoresis (SPE), or UPE were compared to diagnoses of incident M-component related diseases acquired from Danish health registers. UPE and sFLC-ratio equally improved diagnostic values when combined with SPE (sensitivity: SPE and UPE: 95.6 (90.6–98.4); SPE and sFLC-ratio: 95.1 (90.2–98.0)). The addition of the sFLC-ratio to SPE resulted in the identification of 13 patients with MGUS, light chain disease and amyloidosis, which was in line with the addition of UPE to SPE. The number of false-positive tests was UPE and SPE: 364 (11%) and sFLC-ratio and SPE: 677(19%). Expanding sFLC-ratio reference range to 0.26–4.32 resulted in a significant reduction in false positives *n* = 226 (6%) without loss of patients with clinical plasma cell dyscrasias. sFLC-ratio improves the diagnostic value of SPE in GP. However, due to low specificity and a large number of false positives, expanded cut-off values should be considered.

## 1. Introduction

Multiple Myeloma (MM) is rare in general practice (GP) consultations; however, many patients in the GP present with non-specific symptoms that may raise a suspicion of MM [1]. Over the last decades, the overall survival of MM has been rising owing to improvements in treatment and a focus on an early diagnosis [2]. The diagnosis is based on a clinical evaluation and laboratory tests, including the measurement of monoclonal immunoglobulins (M-protein) by serum protein electrophoresis (SPE) and urine protein electrophoresis (UPE). Unfortunately, collection of urine samples is challenging with compliance of 20–40% [3,4,5]; thus, the more convenient measurement of the serum-free light chain (sFLC) ratio was recommended for MM diagnosis by the International Myeloma Working Group [6]. 

The sFLC ratio has been evaluated in selected cohorts with encouraging results, especially in patients with light chain disease, low-secretory disease and amyloidosis [7,8,9,10]. The reported diagnostic values have been very high, with sensitivities and specificities between 96% and 100% [11,12,13,14,15,16,17,18]; however, these results were established in second-line settings or mixed cohorts of patients already selected based on suspicion of MM. So far, the sFLC ratio has not been tested in a first-line setting. 

As the gatekeepers of the health care system, general practitioners (GPs) have to identify at-risk patients in an unselected low-prevalence group to secure an optimised flow into the more specialised health services [1,19]. In this unselected group of patients, the majority will have unspecific or vague symptoms that potentially could represent MM [20]. Therefore, the GPs screen their patients typically by a battery of blood tests to select the right patients for the right ‘track’. Included in the battery of tests are SMP and sometimes UMP but not the sFLC. Uniquely to the Central Denmark Region, sFLC has been available for GPs since 2008. However, consequently, there is a large risk of unnecessary referrals to specialised haematological clinics based on false-positive test results [1,19,20]. Hence, we analysed data from an unselected patient cohort in general practice to investigate (1) whether the sFLC ratio improved diagnostic values when combined with SPE, (2) whether the sFLC ratio could supersede UPE and (3) how different sFLC ratio cut-offs affected the diagnostic values.

## 2. Materials and Methods

### 2.1. Patients

In a retrospective cohort study, patients were identified in the clinical laboratory information system (LABKA) database, which contains all blood test results performed in the Central Region of Denmark. Patients were included if their SPE, UPE, or sFLC ratio was requested by a GP between 18 May 2008 and 31 December 2013 and if the SPE, UPE, or sFLC ratio had not been requested for two years prior to the study period. Patients were excluded if no proper sample was delivered, received, or analysed. Further, patients with known MM, monoclonal immunoglobulin of unknown significance (MGUS), amyloidosis, plasmacytoma, heavy chain disease, or Waldenström macroglobulinemia prior to inclusion were excluded just as patients without a Danish citizenship number. All citizens in Denmark are assigned a citizenship number, which is a unique 10-digit civil personal registration number. The citizenship number is used throughout all the Danish administrative registries and clinical databases and allows unambiguous linking and tracking of all patients. The LABKA database reached the full cover of the Central Denmark Region in 2011. From 2008 to 2011, the database contains blood tests from the most populous counties in the region.

Based on the cohort identified in LABKA, information was extracted from the National Myeloma Registry to identify patients with the monoclonal disease (MM, MGUS, plasmacytoma, amyloidosis, heavy chain disease or Waldenström macroglobulinemia). Patients in the cohort were thus categorised as diseased (monoclonal disease: MCD) or non-diseased (non-MCD). The National Myeloma Registry contains data from all newly diagnosed patients from the hematologic departments in Denmark. The categorisation was verified in the Danish National Patient Registry (DNPR). The DNPR is a registry of all admissions to Danish hospitals containing information on discharge diagnoses, admission and discharge date. The diagnosis coding is based on the International Classification of Diseases and Related Health Problems (ICD-10). All discharge diagnoses between 1 January 1977 and 31 December 2014 were retrieved. To secure the completeness and transparency of the study, the items in ‘Standard for Reporting Diagnostic Accuracy Studies’ (STARD) were followed according to the STARD 2015 [21].

### 2.2. Laboratory Methods

sFLC ratio concentrations were measured using the κ and λ serum FLC assays from The Binding Site Group Ltd on Penta 400 (Horiba) in the period 2008–2010 and since 2011 on the SPAplus (The Binding Site) and Architect C16000 (Abbott). Reference ranges were κ: 3.3–19.4 mg/L, λ: 5.7–26.3 mg/L and FLC ratio: 0.26–1.65 [11]. To adjust for renal impairment, the Freelite reference range for renal disease [22,23] was applied if the creatinine level was >100 µg/L. SPE was measured by capillary electrophoresis and immune-typed using Capillarys (Sebia) and UPE was performed on the Hydrasys (Sebia), which was also used for immune-typing of urine and selected serum samples.

### 2.3. Statistics

Data from LABKA, the National Myeloma Registry and DNPR were linked at an individual level using the citizenship number. Baseline characteristics were compared between MCD and non-MCD patients using the Chi-squared, exact T-test, or the Kruskal–Wallis tests. To visualise the distribution of data as continuous data, κ, λ and sFLC ratios were plotted by kernel density plots. To demonstrate the distribution around the clinical cut-offs, extreme values were given the following values: κ: 50; λ: 50 and sFLC ratio: 6.

The diagnostic value of SPE, UPE and sFLC ratio was estimated by sensitivity, specificity, positive predictive value (PPV) and negative predictive value (NPV). For patients with more than one test result, the diagnostic value was initially estimated using both the first sample and later samples if these were more than three months apart. This was done to investigate if any eventual shift in test results from normal to abnormal (or vice versa) would influence the calculations. However, we found no significant difference in any of the estimated diagnostic measures; therefore, only the first sample was used for further analysis.

As it is well-described that paraproteinemia can be detected in the blood years in advance of detection of a given disease [24,25], we decided to evaluate whether MCD is present or not at any time after the sample was drawn.

Combinations of diagnostic tests were then evaluated and the diagnostic values were compared. When comparing combinations of diagnostic tests, each individual test was evaluated only in the subgroup of patients who had both or all three tests performed within six days. Comorbidity was registered based on information from the DNPR. All tests were two-sided and a *p*-value < 0.05 was considered significant. The analyses were performed using the statistical software STATA, version 14.0.

## 3. Results

### 3.1. Patients

A total of 13,999 patients were identified in the LABKA database, of which 190 patients were excluded (173 had MCD prior to the study period, 13 patients had a non-valid test result, 4 patients did not have a Danish citizenship number) (Figure 1). This left 13,210 patients to be included in the final study cohort.

Baseline characteristics of the study cohort are demonstrated in Table 1. sFLC ratio was measured in 3742 patients (28%). SPE was measured in 12,950 patients (98%), whereas 3373 patients (26%) had a UPE measurement. MCD was diagnosed in 294 (2%) of the patients. The total cohort had a median follow-up time of 34 months (10% percentile: 15 months; 90% percentile: 67 months).

### 3.2. sFLC Ratio Test Results

Density plots of test values for κ, λ and sFLC ratios are visualised in Figure 2. As expected, the majority of non-MCD patients fell within the reference ranges (κ: 62%, λ: 85% and sFLC ratio: 84%). However, a large proportion of non-MCD patients had κ levels slightly above the upper reference limit (Figure 2). This apparent skewness resulted in a right-shift in the non-MCD sFLC ratio (2.5–97.5 percentile interval 0.42–4.32). Figure 2 also illustrates that even for the patients with MCD, the largest fraction had values within the established reference ranges for κ, λ and sFLC ratio.

### 3.3. Diagnostic Value of Individual and Combined Tests

The diagnostic values of the tests are shown in Table 2. As a stand-alone test, SPE had an overall sensitivity of 89.7% and a specificity of 95.9%. The values for UPE and sFLC ratio were lower, with UPE showing the highest specificity and the sFLC ratio having the highest sensitivity at the standard cut-off. The sFLC ratio stands out with 16% false-positive tests. These overall data, however, do not take into account that UPE and sFLC ratios are considered especially useful for patient subgroups such as light chain disease.

Therefore, we calculated diagnostic values by combining SPE with each of UPE and sFLC ratios and considering a positive test result if any of the tests were positive. The addition of either UPE or sFLC ratio to SPE both resulted in an increased sensitivity to >95%; however, the number of false-positive tests adding sFLC ratio (19%) resulted in a significantly lower specificity as compared to UPE. In the subgroup of patients with all three tests performed (*n* = 620), no significant gain in sensitivity was observed by adding the sFLC ratio to UPE and SPE (not shown). 

When combining SPE and sFLC ratio, the number of false-negative tests decreased from 20 to 7, identifying additionally 13 patients. The basic characteristics of these patients are presented in Table 3. Of the 13 patients, 2 were diagnosed with light chain MM, 2 were diagnosed with amyloidosis and 1 patient was diagnosed with Waldenström macroglobulinemia. The remaining 8 patients were diagnosed with MGUS.

### 3.4. Clinical Cut-Offs

Based on our initial observations in the non-MCD group, we questioned whether the diagnostic cut-offs (sFLC ratio: 0.26–1.65) were applicable in the primary care setting. Changing the upper limit to the 97.5 percentile identified in our non-MCD cohort (sFLC ratio: 4.32), the number of false positives decreased from 19 to 6% (Table 2). Importantly, this change in cut-off still identified 10 out of the 13 patients who were additionally identified by applying the combination of sFLC ratio and SPE, and these three non-diagnosed patients all had light-chain MGUS (Table 3). Further increasing the upper limit to 7.0 as suggested by Heanley et al. [26] or 10.0 as suggested by the European Myeloma Network [27] resulted only in minor reductions in false positives (Table 2) and the miss of a patient with Waldenström macroglobulinemia (Table 3).

### 3.5. Comorbidity

We investigated whether comorbidity was related to a large number of false-positive sFLC ratio tests. Comorbidity was evaluated prior to the sample being drawn and three months after. Results are presented in Table 4. As expected, we found that renal disease and cancer prior to the time of testing were associated with an abnormal sFLC ratio. We did not find other associations between common comorbidities and an abnormal sFLC ratio. The median follow-up time for patients with a false positive sFLC ratio was 28 months (10% percentile: 16 months; 90% percentile: 68 months). After 6 months, 21 patients out of 586 patients (3.6%) were deceased. This is in line with the findings in patients with a true negative sFLC ratio. Here, a median follow-up time of 33 months (10% percentile: 14 months; 90% percentile: 62 months) was observed; further, 92 patients out of 3012 patients (3.1%) were deceased.

## 4. Discussion

In this retrospective register-based cohort study based on 13,210 patients, we evaluated the added diagnostic value of the sFLC ratio when applied in general practice consultations.

The sFLC ratio has been fully incorporated in the hematologic work-up for several plasma cell dyscrasias and the diagnostic values have been impressive in the second-line setting [11,12,13,14,15,16,17]. However, whether the sFLC ratio can be used in the primary setting as a screening biomarker for MM and other plasma cell dyscrasias has not been explored. 

Since 2008, the sFLC ratio has been available for GPs in the Central Denmark Region if MM was suspected, though not part of local diagnostic guidelines. Using these unique data, we confirmed that the sFLC ratio, when added to SPE, increased sensitivity to more than 95% and that the sensitivity of the sFLC ratio is in line with UPE when evaluated as a supplement to SPE. The sensitivity of the sFLC ratio was also non-inferior to UPE in the relatively few (*n* = 620) samples in which all three tests had been requested. 

Furthermore, the replacement of UPE with sFLC ratio will probably result in improved overall detection of positive cases due to the low compliance generally obtained for urine UPE (2–4). Even though UPE was part of the recommendations for screening for plasma cell dyscrasia in the study period, only 25% (3373 out of 13,210) of our patients had a UPE measurement. The sFLC ratio was not part of the recommendations for screening for MM in our region during the study period. However, since the sFLC ratio can be performed on the same sample as SPE, the number of tests is expected to approximate 100% of patients if the test is included in local guidelines.

Combining SPE with sFLC ratio resulted in an additional identification of 13 patients who were not identified by SPE alone, mainly MGUS, but as expected, also patients with light chain disease and amyloidosis. However, this improvement in sensitivity came with a very high increase in false-positive tests from five to nineteen per cent. Such poor test specificity in a low prevalence setting results in unnecessary anxiety among patients. Further, it may place a heavy burden on the specialised health care system when patients with abnormal results are referred for further haematological investigation. In Denmark, the sFLC has now been recommended as a first-line screening test in general practice. Therefore, choosing the proper clinical cut-off is of utmost importance for the individual patient and from a cost-benefit point of view. Other countries may choose other strategies based on local socio-economic conditions.

The high number of false-positive tests could be explained by several factors. Firstly, we defined only patients with MCD as having a disease, well aware that other diseases can affect the sFLC ratio. We, therefore, used information from the Danish national registers [28] to explore the role of comorbidities. As expected, renal disease and cancer prior to the time of testing were associated with an abnormal sFLC ratio. However, this cannot explain the high number of false positives. We did not find other associations between common comorbidities and an abnormal sFLC ratio, although previous studies have demonstrated elevated levels of κ and λ in inflammatory diseases affecting B lymphocytes, including connective tissue diseases and chronic liver diseases [29,30,31].

A second explanation for the high number of false positives could be the often-slow progression of plasma cell dyscrasias that may be detected years before diagnosis [24,25]. Unfortunately, our data remains immature to pursue that hypothesis since we have only a median follow-up time of 36 months. 

The most meaningful explanation for the many false positives is the use of non-optimal decision limits. We used the standard reference range by Katzmann et al. that was established based on blood samples from 285 healthy blood donors [11]. This reference range may, however, not be applicable in all settings. Firstly, our non-diseased patients cannot be considered healthy and they represent a true reflection of an unselected cohort in primary care with a non-comparable lifestyle to blood donors and with putative comorbidities. Secondly, it is well known that the choice of analytical equipment and batch of reagents may influence the sFLC ratio [32].

As shown in Figure 2, a large proportion of patients without MCD had κ levels slightly above the upper reference limit, resulting in a right-shift in the sFLC ratio (2.5–97.5 percentile interval 0.42–4.32). If the upper level of 4.32 were applied in our dataset, the number of patients with false-positive test results would decrease significantly from 586 (16%) to 79 (2%) (Table 2). These cut-offs identified 10 out of the 13 MCD patients and the 3 missed patients all had MGUS. The European Myeloma Network recommends further work-up for patients with light chain MGUS but only if the sFLC ratio is very low or very high (e.g., 0.1–10) [27]. By applying these cut-offs, one patient with Waldenström macroglobulinemia would not be identified, but the number of false-positive results would further decrease to 23 (1%). 

The major strength of this study is that it is the first to evaluate sFLC in a large number of unselected patients. The magnitude of our dataset gives robustness to our results that is indisputable. Furthermore, according to the Danish registries’ quality and the Danish health care system, data from different registries could be connected at an individual level. This allowed us to conduct a population-based study including all patients tested for sFLC in a well-defined geographical region, which reduces the risk of selection bias and, more importantly, increases the generalizability of our results. However, there are also limitations to consider. Firstly, we did not have information on why the test was evaluated in the first place. Therefore, patients without any reasonable suspicion of MM could be included in the cohort. However, as we wanted to test real-life data and as GPs often test patients by a pre-specified battery of blood tests, we do accept this in our study. Secondly, we did not have information on comorbidity that was diagnosed and treated at the GP, and, therefore, we cannot rule out that this could affect the sFLC.

## 5. Conclusions

In conclusion, our findings support that the sFLC ratio can replace UPE when patients are suspected of myeloma in general practice. Local instrument-dependent adjustment of reference ranges/decision limits is important to avoid an unnecessarily high number of false-positive tests.

## Figures and Tables

**Figure 1 cancers-14-02930-f001:**
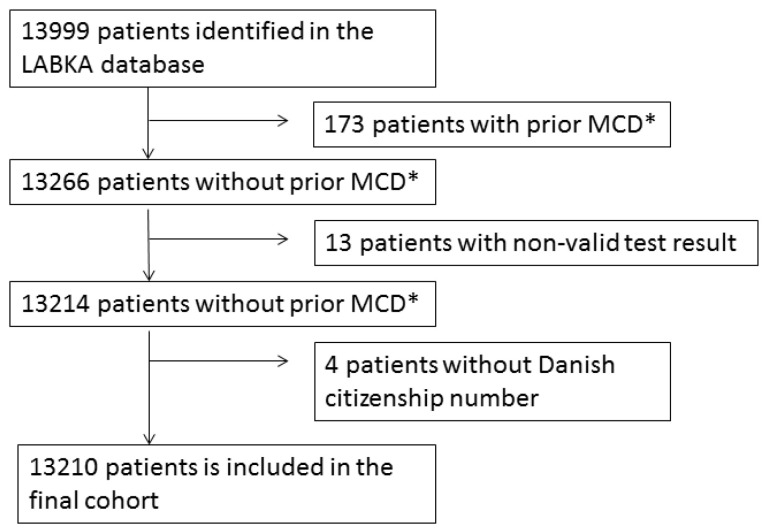
Flowchart.* MCD: monoclonal disease that includes Myelomatosis, MGUS, plasmacytoma, amyloidosis, heavy chain disease and Waldenstrom macroglobulinemia.

**Figure 2 cancers-14-02930-f002:**
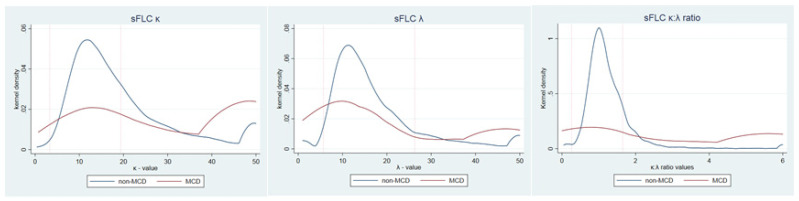
Kernel density of sFLC κ, sFLC λ and the sFLCκ:λ ratio. Kernel density plot depicts the density for sFLC κ, sFLC λ and sFLC κ:λ ratio values for patients with MCD (red line) and patients without MCD (blue line).

**Table 1 cancers-14-02930-t001:** Patients characteristics.

Patients Characteristics	Total	Patients without MCD	Patients with MCD
Total, *n*	13,210	12,916	294
Age			
Years, Median (range)	64 (1–106)	64 (1–106)	70 (39–97)
Sex			
Female, *n* (%)	7961 (60)	7841 (61)	120 (41)
Male, *n* (%)	5249 (40)	5075 (39)	174 (59)
Creatinine, µmol/L, Median (range)	71 (40–280)	71 (41–189)	76 (40–280)
Number of patients for each analysis, *n* (%)			
sFLC κ:λ	3742 (28)	3595 (28)	144 (49)
SPE	12,950 (98)	12,658 (98)	292 (99)
UPE	3373 (26)	3237 (25)	136 (46)
Combinations of analyses, *n* (%)			
SPE	6717 (51)	6619 (51)	98 (33)
SPE and UPE	2649 (20)	2598 (20)	51 (17)
UPE	102 (<1)	101 (1)	1(<1)
SPE and FLC ratio	2964 (22)	2905 (22)	59 (20)
UPE and FLC ratio	2 (<1)	2 (<1)	0
FLC ratio	156 (<1)	155 (1)	1 (<1)
SPE, UPE and FLC ratio	620 (5)	536 (4)	84 (29)

MCD, Monoclonal immunoglobulin disease (Myeloma, MGUS, plasmacytoma, amyloidosis, Waldenström macroglobulinemia); sFLC κ:λ, serum-free light chain κ:λ ratio; SPE, serum monoclonal protein; UPE, urine monoclonal protein.

**Table 2 cancers-14-02930-t002:** Diagnostic values.

Diagnostic Tests	Total (N)	MCD (N)	Sensitivity	Specificity	PPV	NPV	False Positive N (%)	False Negative N (%)
Individual tests								
SPE	12,950	292	89.7 (85.7–93.0)	95.9 (95.6–96.3)	33.7 (30.4–37.2)	99.8 (99.6–99.8)	515 (4)	30 (<1)
UPE	3373	136	50.7 (42.0–59.4)	92.3 (91.3–93.2)	21.6 (17.2–26.6)	97.8 (97.2–98.3)	250 (7)	67 (2)
sFLC κ:λ (0.26–1.65)	3742	144	71.5 (63.4–78.7)	83.7 (82.5–84.9)	14.9 (12.4–17.8)	98.7 (98.2–99.0)	586 (16)	41 (1)
sFLC κ:λ (0.26–4.32)	3742	144	51.4 (42.9–59.8)	97.8 (97.3–98.3)	48.4 (40.2–56.6)	98.0 (97.5–98.5)	79 (2)	70 (2)
sFLC κ:λ (0.26–7.0)	3742	144	43.8 (35.5–52.3)	98.4 (98.0–98.8)	52.9 (43.6–62.2)	97.8 (97.2–98.2)	56 (1)	81 (2)
sFLC κ:λ (0.1–10.0)	3742	144	34.7 (27.0–43.1)	99.4 (99.0–99.6)	68.5 (56.6–78.9)	97.4 (96.9–97.9)	23 (1)	93 (2)
Combination of test * SPE and UPE ^a^	3269	135	95.6 (90.6–98.4)	88.4 (87.2–89.5)	26.2 (22.3–30.3)	99.8 (99.5–99.9)	364 (11)	6 (<1)
SPE and sFLC κ:λ (0.26–1.65) ^b^	3584	143	95.1 (90.2–98.0)	80.3 (79.0–81.6)	16.7 (14.2–19.5)	99.7 (99.5–99.9)	677 (19)	7 (<1)
SPE ^a^	3269	135	87.4 (80.6–92.5)	95.0 (94.2–95.7)	42.9 (37.0–49.0)	99.4 (99.1–99.7)	157 (5)	17 (3)
SPE ^b^	3584	143	86.0 (79.2–91.2)	95.1 (94.3–95.8)	42.1 (36.4–48.0)	99.4 (99.1–99.6)	169 (5)	20 (1)
UPE ^a^	3269	135	50.4 (41.6–59.1)	92.2 (91.2–93.1)	21.8 (17.3–26.8)	97.7 (97.1–98.2)	244 (7)	67 (2)
sFLC κ:λ (0.26–1.65) ^b^	3584	143	71.3 (63.2–78.6)	83.3 (82.1–84.6)	15.1 (12.5–18.0)	98.6 (98.1–99.0)	573 (16)	41 (7)
Alternative reference ranges for sFLC κ:λ SPE + sFLC κ:λ (0.26–4.32) ^b^	3584	143	93.0 (87.5–96.6)	93.6 (92.7–94.4)	37.7 (32.6–43.0)	99.7 (99.4–99.9)	229 (6)	10 (<1)
FLC- κ:λ ^b^	3584	143	51.0 (42.6–59.5)	97.8 (97.2–98.2)	48.7 (40.4–57.0)	98.0 (97.4–98.4)	77 (2)	70 (2)
SPE + sFLC κ:λ (0.26–7.00) ^b^	3584	143	92.3 (86.7–96.1)	94.0 (93.2–94.8)	39.2 (33.9–44.6)	99.7 (99.4–99.8)	205 (6)	11 (<1)
sFLC κ:λ ^b^	3584	143	43.4 (35.1–51.9)	98.4 (98.0–98.8)	53.4 (44.0–62.8)	97.7 (97.1–98.1)	54 (2)	81 (2)
SPE + sFLC κ:λ (0.1–10) ^b^	3584	143	92.3 (86.7–96.1)	94.8 (94.0–95.5)	42.3 (36.8–48.0)	99.7 (99.4–99.8)	180 (5)	11 (<1)
sFLC κ:λ ^b^	3584	143	34.3 (26.5–47.7)	99.3 (99.0–99.6)	68.1 (56.0–78.6)	97.3 (96.7–97.8)	23 (<1)	94 (3)

* When testing and comparing combinations of diagnostic tests, only patients with matched samples were included. The combined result was considered positive if one test was outside the reference interval. For comparison reasons, each individual test was also evaluated in the subgroups of patients with a combination of tests available. ^a^ tested in the subgroup of patients with measurements of SPE and UPE available. ^b^ tested in the subgroup of patients with measurements of SPE and sFLC κ:λ available. MCD, Monoclonal immunoglobulin disease (Myelomatosis, MGUS, plasmacytoma, amyloidosis, Waldenström macroglobulinemia, Heavy Chain Disease); NPV, negative predictive value, PPV, positive predictive value; sFLC κ:λ, serum-free light chain κ:λ ratio; SPE, serum monoclonal protein; UMP, urine monoclonal protein.

**Table 3 cancers-14-02930-t003:** Patients not identified by SMP.

Patient	Diagnosis	κ	λ	sFLC κ:λ	UPE	Days from Sample until Diagnosis
1	MM	1728	1	1728	N/A	2
2	MM	13	386	0.03	pos	150
3	Amyloidosis	681	11	61.9	N/A	35
4	Amyloidosis *	64	859	0.07	pos	47
5	WM *	33	6	5.48	pos	65
6	MGUS	27	15	1.8 **	neg	34
7	MGUS	1350	6	225	pos	351
8	MGUS	559	18	31.1	pos	9
9	MGUS*	155	36	4.31 **	pos	890
10	MGUS	490	39	12.6	pos	17
11	MGUS	187	7	26.7	pos	12
12	MGUS*	22	13	1.69 **	pos	20
13	MGUS	13	287	0.05	pos	2

Information on the 13 patients identified by sFLC and with a negative M-component in plasma (SPE). All patients had creatinine within the normal range. sFLC κ:λ, serum-free light chain κ:λ ratio; UPE, urine monoclonal protein; N/A: not assed; WM: Waldenström macroglobulinemia.* serum monoclonal protein appears during the follow-up period. ** Due to the low sFLC κ:λ ratio, the patient would not be identified if cut-off values were changed.

**Table 4 cancers-14-02930-t004:** Comorbidity.

Comorbidity	Comorbidity Prior to Blood Test							Comorbidity 3 Months after Blood Test						
*n* = 3742		MCD			Non-MCD				MCD			Non-MCD		
	Total	sFLC Abnomal	sFLC Normal	*p*-Value *	sFLC Abnormal	sFLC Normal	*p*-Value *	Total	sFLC Abnormal	sFLC Normal	*p*-Value *	sFLC Abnormal	sFLC Normal	*p*-Value *
Myocardial infarction	152	4	0	NA	24	124	0.98	166	4	0	NA	25	137	0.76
Congestive heart failure	398	9	2	NA	62	325	0.88	438	10	3	NA	72	353	0.70
Peripheral vascular	386	5	4	NA	56	321	0.43	437	7	5	0.32	63	362	0.38
Cerebrovascular	205	5	0	NA	34	166	0.78	232	5	0	NA	38	189	0.85
CPD	290	7	2	NA	45	236	0.90	344	7	2	NA	49	286	0.39
Diabetes	210	5	0	NA	42	163	0.09	251	5	0	NA	50	196	0.08
Diabetes with end-organ damage	68	2	0	NA	12	54	0.84	79	2	0	NA	14	63	0.65
Renal disease	82	0	2	NA	25	55	<0.001	118	2	2	NA	32	82	0.001
Mild liver disease	15	1	0	NA	2	12	NA	24	1	0	NA	2	21	NA
Ulcer disease	51	2	1	NA	7	41	1.00	76	3	1	NA	8	64	0.26
Connective tissue disease	50	1	0	NA	11	48	0.24	67	8	1	NA	14	52	0.27
Any tumour	204	5	1	NA	53	145	<0.001	396	8	1	NA	77	310	0.04
Leukaemia	2	0	0		1	1	NA	10	0	0		4	6	NA
Lymphoma	0							4	1	0	NA	0	3	NA
Metastatic solid tumor	57	2	1	NA	14	40	0.05	907	17	8	0.64	138	744	0.55

* the number of patients with the condition are reported in this table and compared to the number of patients without the condition by the the Pearson chi2 or exact *T*-test. CPD: Chronic pulmonary disease; MCD: Monoclonal immunoglobulin disease (Myeloma, MGUS, plasmacytoma, amyeloidosis, Mb Waldenström); NA: non assed; sFLC, serum free light chain κ:λ ratio.

## Data Availability

In this study, publicly archived datasets were achieved from the Danish National Myeloma Registry (www.danishhealthdata.com) accessed on 1 January 2019, the Danish National Patient Registry (https://sundhedsdatastyrelsen.dk/da/english/health_data_and_registers/national_health_registers) accessed on 1 January 2019, and the clinical laboratory information system (LABKA) www.auh.dk/om-auh/afdelinger/blodprover-og-biokemi accessed on 1 January 2019.

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
