# Peer review of "A Cohort Study of Free Light Chain Ratio in Combination with Serum Protein Electrophoresis as a First-Line Test in General Practice"

_cancers, 2022, doi:10.3390/cancers14122930_

Round 1
Reviewer 1 Report
This report describes the merits and drawbacks of sFLC testing in a very large population cohort.
The major flaw is the lack of baseline cause for sending the test. This is not highlighted in the text, and not discussed. were 30% tests were sent because of MM suspicion? So there is a significant bias in the preliminary patient selection. Eventually only 2% were diagnosed with MCD- this is also extremely low for MGUS only (how do the authors explain this as well?), as probably the FLC was sent to median age of 60-70. Perhaps limit the investigations to ages 40 and up? (why would FLC be sent to a 1 year old? certainly not because of MM suspicion) This bias is the major bias of this work. Need to find a way to overcome it.
In addition the authors are encourged to add limitations of the study to the discussion.
Another point would be - NNT. What are the numbers needed in order to find 4 MM/ Amyloidosis patients at screening (and 9 MGUSs... is this important?)? e.g. - (and please discuss) - there is no point for performing FLC alone as a screening test.
minor remarks:
Please change danish terms to the common English used - i.e. Amyloidose to Amyloidosis and Mb Waldenstrom to WM or LPL.
Table 1 - add baseline creatinine levels and possibly other cancer status.
Figure 1 MCD is NOT myelomatosis. change this term.
Figure 2 - How come for MCD the max ratio is 6? If there are MM or Amyloidosis patients found- their ratios are expected to be much higher?
Reviewer 2 Report
This paper shows the interest of measuring serum free light chain ratio to confirm the diagnosis significance of an abnormal serum protein electrophoresis in general practice setting.
Interestingly, it confirms that in this low prevalence setting, an extension of sFLC ratio reference range reduces false positive cases while affecting very little the sensitivity of the test.
Results are sound and clearly presented. However, I have a few questions:
- Different devices were used to measured sFLC during the study ( Penta 400, SPAplus and Architect C16000). Has it been verified that the results obtained with these different devices are absolutely identical ?
- Sensitivities calculated for the combination of SPE and UPE or SPE and sFLC ratio are slightly different between the abstract and table 2 :
SPE + UPE : abstract 95.2 % ( 88.3 – 98.7) table 2 : 95.6% ( 90.6 – 98.4)
SPE + sFLC ratio : abstract 94.0 % ( 86.7 – 98.0) table 2 : no identical value whatever the chosen reference range.
Results show that addition of FLC measurement to SPE increase sensitivity for monoclonal disease by 3.3 %, choosing a reference range between 0.26 and 4.32 for sFLC ratio to avoid false positive results. This is certainly useful, but also greatly increases the cost of the evaluation given the very large number of patients that would need to be investigated. This aspect should also be discussed.
Finally, I have a concern about the title of the paper. It is a little misleading as it describes the value of free light chain ratio to replace urine protein electrophoresis, but not serum protein electrophoresis which remains the first line test.
Reviewer 3 Report
The authors analyzed the value of sFLC examination in the GP setting and its potential to replace a costly, time-consuming, and quite difficult for patients UPE test in patients suspected of plasma cell dyscrasias.
As a hematologist working very closely with multiple myeloma patients, I believe this is a vital aspect and could improve the detectability of MM patients, which is still too low. I find the study design appropriate and well thought out, and the presented data checked on a vast number of patients to be accurate. The statistical analysis seems to be on point.
I especially liked the analysis of extending the sFLC ratio for broader limits and its influence on the number of false-positive results. The authors' suggestions seem to support my observations. They should be considered, as the group of patients analyzed was much bigger than the one used to establish the reference ranges.
The language used in the article is also correct; however, I identified quite a lot of minor spelling and interpunction mistakes. Therefore, I would suggest the authors proofread the article with the help of a native speaker or spellchecking software to improve the overall merit of the article.
Apart from that, I do recommend publishing this article.
Round 2
Reviewer 1 Report
The authors have responded to all quaries